# The Functions and Phenotypes of Microglia in Alzheimer’s Disease

**DOI:** 10.3390/cells12081207

**Published:** 2023-04-21

**Authors:** Risako Fujikawa, Makoto Tsuda

**Affiliations:** 1Department of Molecular and System Pharmacology, Graduate School of Pharmaceutical Sciences, Kyushu University, Fukuoka 812-8582, Japan; 2Kyushu University Institute for Advanced Study, Fukuoka 819-0395, Japan

**Keywords:** Alzheimer’s disease, microglia, amyloid beta, tau, brain, spinal cord, pain, DAMs, CD11c, ApoE, TREM2, monocyte

## Abstract

Alzheimer’s disease (AD) is the most prevalent neurodegenerative disease worldwide, but therapeutic strategies to slow down AD pathology and symptoms have not yet been successful. While attention has been focused on neurodegeneration in AD pathogenesis, recent decades have provided evidence of the importance of microglia, and resident immune cells in the central nervous system. In addition, new technologies, including single-cell RNA sequencing, have revealed heterogeneous cell states of microglia in AD. In this review, we systematically summarize the microglial response to amyloid-β and tau tangles, and the risk factor genes expressed in microglia. Furthermore, we discuss the characteristics of protective microglia that appear during AD pathology and the relationship between AD and microglia-induced inflammation during chronic pain. Understanding the diverse roles of microglia will help identify new therapeutic strategies for AD.

## 1. Introduction

Alzheimer’s disease (AD) is the most common cause of dementia, memory loss, and other cognitive disabilities that interfere with daily life [1]. The number of patients with dementia is estimated to reach 75 million by 2030, and their burden may cause healthcare systems to collapse [2,3]. Although aggregation of amyloid-β (Aβ) and tau proteins may play crucial roles in the progression of AD, candidate drugs targeting Aβ or tau have failed to reach efficacy clinical endpoints for improving cognitive function [4].

As new therapeutic targets for Alzheimer’s disease are being explored, microglia, which are resident immune cells of the central nervous system (CNS), are attracting increasing attention. Microglia has been reported to contribute to AD pathology through a variety of mechanisms, including cytokine production and Aβ clearance [5]. The recent introduction of single-cell or single-nucleus RNA sequencing (scRNA-seq or snRNA-seq, respectively) analyses has greatly advanced the knowledge regarding microglia in AD pathogenesis [6]. In addition, genome-wide association studies (GWASs) have reported a list of risk factor genes expressed in microglia, thus suggesting additional potential cellular targets for AD treatment [7].

In this review, we have summarized the risk genes expressed in microglia and the response of microglia to Aβ and tau to argue the importance of microglia in AD. Moreover, we have summarized the relationship between chronic pain, microglia, and AD to facilitate a comprehensive understanding of AD, including its complications. Microglial phenotypes and candidate drugs have also been discussed to explore future therapeutic targets for AD. 

## 2. Microglia

Microglia are critically involved in many physiological and pathological brain processes throughout life (pre- and post-natal development as well as aging). Individual microglial cells are estimated to be in contact with 2–100 neurons depending on the neuronal density [8]. Microglia are influenced by various factors, including interactions with other cells. For example, interleukin-34 (IL-34), derived from neurons or retinal ganglion cells, and colony-stimulating factor 1 (CSF-1), derived from neurons or astrocytes, bind to microglial CSF-1 receptors which are crucial for microglial development and survival [9,10]. Furthermore, the interaction of C-X3-C motif chemokine ligand 1 (CX3CL1) released from neurons with its receptor CX3C receptor 1 (CX3CR1) on microglia influences synaptic maturation and plasticity [11]. Through microglial IL-1 receptor-like 1, IL-33 from astrocytes promotes microglial synapse engulfment [12]. C-C motif chemokine 2 (CCL2) and C-X-C motif chemokine ligand 10 (CXCL10) modulate microglial recruitment to injured areas [13].

Morphological changes in microglia are one of the features observed in CNS diseases [14]. Microglia in the healthy adult brain usually exhibits a ramified morphology with complex dendrites and small somas. However, the number, length, and complexity of the branching of these processes can change under various pathological conditions, including stress or injury [15,16]. In mouse models of AD, microglia with decreased complexity, called ameboid microglia, have been observed [17]. In the brains of human patients with AD, the number of microglia is not affected, but the ramified population is decreased [18]. Aβ removal by immunotherapy in patients with AD leads to increased ramified microglia, implying that the cells retain plasticity in the aged AD brain [18]. It is paramount to examine changes in microglial function after drug treatment for AD.

Microglia modulate brain activity through a variety of functions, such as phagocytosis and cytokine release. Microglia are the brain’s main phagocytes and protect the brain by phagocytosing bacteria, aggregated proteins, and cellular debris [19,20]. Phagocytic elimination of synapses by microglia affects the balance between excitatory and inhibitory synaptic transmission (E/I balance) [21]. Disruption of the E/I balance due to the impaired synaptic exclusion by microglia can lead to a variety of problems in brain function, including memory, cognitive function, sleep, movement, social behavior, and thinking [21]. Recent studies have shown that pro-inflammatory cytokines generally promote healing during the acute stages of injury but in the chronic condition of the disease, they turn into a detrimental process, interfering with repair mechanisms and promoting degeneration [22]. Thus, microglia have a variety of functions and act as a “double-edged sword” both protecting against and exacerbating CNS diseases. Elucidating the complex functions of microglia and how they interact with other cells at the molecular and cellular levels may help in the development of new therapies to more precisely intervene in AD progression.

The microglia are altered in CNS diseases, including AD, but very few studies have tracked the microglia in vivo. This is due to the difficulty in biopsies of human brain tissue and the absence of specific positron emission tomography (PET) tracers for microglia [23,24]. A recent study has indicated that extracellular vesicles (EVs) in the plasma may non-invasively assess the inflammatory response of microglia in an experimental stroke model [25]. After inflammation, microglia-derived EVs contain the microglial protein TMEM119, along with an increased expression of the Toll-like receptor 4 (TLR4) co-receptor CD14 [25]. Using this technology for AD would lead to the diagnosis of the disease stage and early treatment.

## 3. Expression of AD Risk Gene in Microglia

Recently, genomic research has suggested that many single-nucleotide polymorphisms (SNPs) are correlated with differential AD risk [26]. SNP variants can occur within gene-coding regions or influence disease risk through promoters and enhancers. It has been reported that approximately two-thirds of new AD-risk SNPs are most highly or exclusively expressed in microglia [26]. For example, a recent study of over 300,000 individuals reported 48 AD-risk SNPs (FDR < 10^−5^), 29 of which were expressed most highly in microglia (60.4%) [27,28]. These data suggested that microglia play a greater role in AD development and progression than previously thought. Several AD risk genes associated with microglia are described below.

The most common genetic risk factor for developing AD is *APOE* which encodes apolipoprotein E. Humans have three variants of the *APOE* gene: ε2, ε3, and ε4. *APOE* genotypes can affect various cellular functions such as Aβ clearance, synaptic integrity, lipid transport, glucose metabolism, blood–brain barrier (BBB) integrity, and mitochondrial regulation [29]. While carrying the ε2 allele is protective against AD, carrying the ε4 allele is an AD risk factor for AD [29,30]. The risk of AD increases from 20% to 90%, and the mean age at onset decreases from 84 to 68 years, with an increasing number of *APOE*-ε4 alleles [31]. ApoE immunoreactivity was observed in dense core plaques as well as in microglial processes around the plaque cores in the brains of 5xFAD mice and patients with AD [32,33]. It has been suggested that ApoE directly interacts with Aβ peptides to form an Aβ/ApoE complex [34]. These complexes are phagocytosed by microglia through interactions with low-density lipoprotein receptor-related protein 1 (LRP1) or triggering receptors expressed on myeloid cells 2 (TREM2) [35,36]. Recent research has shown that microglia-specific ApoE knockout does not alter senile plaque pathogenesis or gene expression, including disease-associated microglia (DAM)-associated genes, but decreases synaptic proteins (post-synaptic marker PSD95 and presynaptic marker synaptophysin) in 5xFAD mice [37]. In the absence of microglial ApoE, astrocyte-derived ApoE may be effective, although further research is required. 

Another AD risk gene is TREM2 which codes for a cell surface transmembrane glycoprotein with a V-immunoglobulin extracellular domain and a cytosolic tail [38]. In the brain, TREM2 is highly expressed by microglia; however, there is still debate as to whether TREM2 is expressed only in a specific subset of microglia in mice and humans [39,40]. Heterozygous rare variants of TREM2 are associated with a significant increase in the risk of AD with an odds ratio similar to that of a single copy of the ApoE ε4 allele [41,42]. TREM2 binds to lipoproteins and apolipoproteins, including ApoE, and Aβ-lipoprotein complexes are efficiently taken up by microglia in a TREM2-dependent manner [43]. The level of TREM2 has been reported to be increased in the brains of AD patients and mouse models [44,45], and TREM2 overexpression is thought to be related to the recruitment of microglia to Aβ plaques [46]. In addition, recent research has demonstrated that TREM2 drives the microglial response to Aβ via spleen tyrosine kinase (SYK)-dependent and -independent pathways in the 5xFAD model [47]. SYK plays a critical role in mounting protective antifungal immune responses downstream of the C-type lectin (CLEC) receptors expressed on innate immune cells. Furthermore, SYK has been identified as a critical kinase that directs signaling and effector functions downstream of TREM2, CD33 (Siglec-3), and CD22 receptors [48,49,50]. It has also been reported that phagocytic clearance of neurotoxic materials by microglia is dependent on SYK in 5xFAD mice [51]. Although TREM2-SYK signaling may be a target for the development of new strategies to treat AD, further research is needed to determine how TREM2 is activated and regulated. In addition, TREM2 is proteolytically cleaved and released into the extracellular space as a soluble variant (sTREM2) that can be measured in cerebrospinal fluid (CSF). As the sTREM2 levels in the CSF increase in the early symptomatic phase of AD, it is a potential biomarker for microglial activity and early-stage AD [52].

CD33 is considered a myeloid-specific immunomodulatory receptor; however, recent research has shown that CD33 is also expressed by microglia in the normal human brain [53]. CD33 is a sialic acid-binding immunoglobulin-like lectin (Siglecs). Siglecs recognize sialic acid residues on glycoproteins and glycolipids and have one or more immunoreceptor tyrosine-based inhibitory motifs (ITIMs) that mediate cell-to-cell interactions that inhibit or limit immune responses [54]. Common variants of CD33 have been reported to be associated with AD [55]. In the AD brain, the number of CD33-immunoreactive microglia is positively correlated with insoluble Aβ42 levels and plaque burden [56]. In addition, CD33 inactivation reduced insoluble Aβ42 levels in APP/PS1 mice [56]. It is possible that the loss of microglial degradative capacity of Aβ in AD can be reversed therapeutically by inhibition of CD33 activity.

## 4. Microglial Responses to Aβ Plaques

Although the underlying cause of AD is not fully understood, Aβ plaques and tau tangles are so far referred to as the key players for disease progression [57]. Under healthy conditions, amyloid precursor protein (APP) is digested by α- and γ-secretase enzymes [58]. This reaction produces soluble polypeptides which can be broken down and recycled. However, when β-secretase works with γ-secretase, insoluble Aβ peptide is produced [58]. Aβ peptide is capable of self-assembling and transforming into soluble oligomeric or insoluble aggregates which are observed in patients with AD, leading to a loss of synapses and neurons [59]. This hypothesis has led to the development of novel Aβ-based therapeutic strategies. In 2021, the FDA approved a new antibody drug for AD called aducanumab which degrades toxic Aβ forms [60]. Contrary to expectations, clinical trials have, however, shown little or no functional improvement in AD patients treated with this drug. Currently, the use of aducanumab is recommended only for AD patients with mild cognitive impairment or dementia [61]. Although many researchers have continued efforts to discover new AD drugs, there is currently still no effective treatment for AD. It is, therefore, paramount to gain a greater understanding of the underlying mechanisms of AD pathogenesis, including the Aβ cascades.

During AD progression, the accumulation of Aβ affects microglia [62]. Aβ induces various microglial responses, including phagocytosis, cytokine release, and proliferation (Figure 1) [63]. Aβ can bind to several receptors on the surface of microglia. Examples include class A scavenger receptor (SR-A1), TLR4, TREM2, the receptor for advanced glycation end products (RAGE), P2X7 receptor, and formyl peptide receptor-like-1 or N-formyl peptide receptor 2 (FPRL1). These receptors mediate the adhesion of microglia to Aβ, leading to phagocytosis [64,65,66,67,68].

Aβ causes microglia to secrete pro-inflammatory cytokines and neurotoxic substances, including tumor necrosis factor-α (TNF-α), IL-1, IL-6, IFN-γ, and reactive oxygen species [69]. First, TNF-α is one of the most important pro-inflammatory cytokines in AD [70]. An in vitro study has shown that Aβ directly stimulates TNF-α production in microglia by activating the transcription factor NF-κB [71]. TNF-α increases the expression of β- and γ-secretases which generate Aβ from APP in vitro [72,73]. In healthy adults, the levels of TNF-α in the cerebrospinal fluid or blood and CNS are very low, but they are significantly elevated in the blood and CNS of patients with AD [74,75]. It has been reported that the risk of AD is lower in patients with rheumatoid arthritis and psoriasis who have been treated with TNF-α blocking agents [76]. IL-1 is also a key factor in AD because this factor leads astrocytes to release α1-antichymotrypsin (ACT) which promotes Aβ deposition [77]. IL-1β regulates the synthesis of APP, secretion of APP from glial cells, and amyloidogenic processing of APP in in vitro experiments [73]. Other microglial inflammatory mediators also contribute to the progression of AD. For example, overexpression of IL-6 induces neuroinflammation and impairs learning and memory functions [78]. In addition, IFN-γ induces lasting microglial priming which includes proliferation and moderate nitric oxide release, resulting in impairment of cognitive function [79]. As discussed above, Aβ and microglia interact with each other to accelerate inflammatory responses and neural damage, leading to the persistent development of AD.

Monocytes were previously thought to be involved in Aβ clearance from brain parenchyma. Monocytes are a type of immune cells that are produced in the bone marrow and travel through the blood to tissues in the body where they become macrophages or dendritic cells. Owing to the difficulty in histologically distinguishing peripherally derived monocytes from resident microglia in human tissues, whether monocytes infiltrate the brain has long been a matter of debate [80]. Using a panel of bone marrow chimeric and adoptive transfer experiments, circulating Ly-6C^high^ C-C motif chemokine receptor 2 (CCR2)^+^ monocytes were shown to be preferentially recruited to the lesioned brain and differentiated into microglia-like cells [81]. However, several studies using bone marrow chimeras generated by head-protected irradiation or chemotherapy-induced myeloablation have indicated that Ly6Chi monocytes are not recruited to the brain parenchyma during AD pathogenesis [82,83]. In addition, despite a nearly complete exchange of resident microglia with peripheral myeloid cells in CD11b-HSVTK mice, there was no significant change in Aβ burden or APP processing [84]. These results suggested that either monocyte do not infiltrate the brain parenchyma in large enough quantities to affect the amount of Aβ in the brain, or that they are not involved in phagocytosis or clearance of Aβ within the brain parenchyma.

## 5. Microglia Responses to Tau Tangles

Tau protein is expressed mainly in neurons, and to a lesser extent oligodendrocytes, and regulates the assembly and stabilization of microtubules [85]. Tau protein undergoes various post-translational modifications, including phosphorylation, glycosylation, acetylation, nitration, methylation, prolyl-isomerization, ubiquitylation, sumoylation, and glycation [86]. Among these modifications, phosphorylation occurs most frequently. Tau phosphorylation is regulated throughout life, but hyperphosphorylation occurs in AD pathology [87]. Tau hyperphosphorylation reduces binding to microtubules and promotes tau aggregation, resulting in impaired neuronal function [88]. Hyperphosphorylated tau tends to accumulate in somato-dendritic compartments which impairs the integrity of microtubules and induces synaptic dysfunction [89].

In a tauopathy mouse model, immunosuppressants inhibited inflammatory responses, attenuated tau pathology, and increased lifespan [90]. Another study suggested that microglia are sufficient to drive tau pathology and correlate with the spread of pathological tau [91]. It has also been demonstrated that tau is phagocytosed by the microglia [92]. Importantly, phosphorylated tau interrupts neuron–microglia interaction [93]. One of the sensors for neuronal signals is the CX3CL1/CX3CR1 axis. CX3CL1 is produced almost exclusively by neurons, and its receptor CX3CR1 is expressed almost exclusively on the surface of microglia [94]. The CX3CL1/CX3CR1 axis is essential for maintaining brain functions, including synaptic integration of adult-born hippocampal granule neurons and regulation of emotional behavior, as demonstrated by in vivo experiments [95]. It has been suggested that the CX3CL1/CX3CR1 axis plays a key role in the phagocytosis of tau by microglia and is affected as AD progresses (Figure 2) [96]. Tau binding to CX3CR1 triggers phagocytosis of tau by microglia, but S396 phosphorylation of tau decreases the binding affinity of this protein to CX3CR1 [96]. In addition, the CX3CL1/CX3CR1 axis may be impaired in the late stages of AD, as the phagocytic capacity of microglia in brain tissue samples from patients with AD is decreased [96]. Furthermore, the CSF and blood levels of CX3CL1 in AD patients are positively associated, suggesting that CX3CL1 may be a potential blood biomarker [97]. Patients with mild cognitive impairment (MCI) and AD have higher plasma levels of CX3CL1 than healthy participants [98,99]. In addition, CX3CL1 levels decreased in severe AD patients [mini mental state examination (MMSE) scores ≤ 14] than those in mild to moderate AD patients (MMSE scores > 14) [99]. Thus, the CX3CL1/CX3CR1 axis plays a key role in AD, and CX3CL1 can be used to diagnose early-phase AD.

## 6. Pain Is a Risk Factor for AD: Possible Involvement of Microglial CNS Inflammation

Patients with AD often have multiple comorbidities. However, there are very few studies that have focused on the interaction between AD and other diseases. Particularly, chronic pain, one of the most commonly observed complications, is associated with increased microglia in the brain and the spinal cord which are involved in pain onset and intensification [100]. Based on a meta-analysis, the prevalence of chronic pain in AD patients was 45.8% [101]. A longitudinal cohort of elders also showed an association between persistent pain and memory decline and dementia [102]. Higher levels of pain interference are associated with a higher probability of developing dementia [103]. Recently, Yamada et al. reported that the dementia risk was higher in participants aged 65–79 years with knee pain and without low back pain than in those without knee and low back pain [HR:1.73 (95% CI:1.11–2.68)] [104]. Another study showed that a history of migraine was significantly associated with AD (OR = 4.22; 95% CI = 1.59–10.42), even after adjustment for confounding and intervening variables [105]. Other conditions, such as fibromyalgia and irritable bowel syndrome (IBS), have also been suggested as possible risk factors for AD [106,107]. The tolerance time in the cold pressor test (4 °C) was lower in patients with AD dementia than in controls [108]. It must be considered that pain may be underestimated in AD patients, as they are not able to communicate and alert to pain compared with cognitively normal patients [109]. In addition, safety issues exist regarding the use of non-steroidal anti-inflammatory drugs, opioids, and adjunctive analgesics in patients with dementia [110]. Further research is needed to provide better methods for pain management in dementia.

In chronic pain conditions, neuroinflammation induced by microglia is a characteristic feature in both humans and animal models [69]. A positron emission tomography (PET) study using [^11^C] PBR28 showed that microglial inflammatory signals were widely elevated in certain cortical regions in patients with fibromyalgia [111]. In different types of chronic pain models, microglial changes in number or morphology have been observed in the spinal cord [112], prefrontal cortex, hippocampus, amygdala [113], nucleus accumbens, thalamus, and somatosensory cortex [114,115]. During these chronic pain states, microglia release pro-inflammatory cytokines, including IL-6, IL-1β, and TNF-α, which cause dysfunction in synaptic remodeling and brain connectivity [100,113,116]. In addition, it has been reported to increase the mRNA expression of IL-1β and TNF-α and accelerate Aβ deposition in 5xFAD model mice [117]. Chronic pain-induced microglial inflammation is associated with AD progression.

Growing evidence suggests that chronic pain may aggravate AD neuropathogenesis through locus coeruleus-noradrenaline (LC-NA)-induced microglial neuroinflammation [116]. LC is involved in various physiological functions, including stress reactions, attention, memory, emotion, and pain modulation [118]. Located in the dorsal habenula, the LC provides descending NAergic inputs to the spinal cord, forming an LC-spinal cord descending pain modulation system [116]. In a rat model of neuropathic pain (chronic pain occurring after a lesion or disease in the somatosensory system), the expression of tyrosine hydroxylase, dopamine β-hydroxylase (DβH), the NA transporter, the α_2_-adrenoceptor, and burst firing were significantly increased in the LC [119,120]. Increased LC-prefrontal cortex (PFC) neurotransmission has been reported in chronic pain models which leads to increased NAergic fiber sprouting and intrinsic excitability in the PFC, mediated in part by α_2_-adrenoreceptors and HCN channels [121]. Microglia respond to NA signaling through α_1_, α_2_, β_1_, and β_2_-adrenergic receptors [122,123]. Morphological changes in microglia induced by social stress were completely blocked by propranolol which is an antagonist of β_1_ and β_2_ adrenergic receptors [124]. In primary rat microglia, NA enhances lipopolysaccharide-induced expression of cyclooxygenase-2 and secretion of prostaglandin E2 via β-adrenoreceptors [125]. These lines of evidence suggest that chronic pain-induced hyperactivity of LC-NAergic neurons may exacerbate AD by causing microglia-induced neuroinflammation. Further studies are needed, as NA dysfunction in the comorbidity of AD and chronic pain is currently still unknown. If the role of microglia in the interaction between pain and AD is clarified, microglia may be targeted for treating both diseases.

## 7. Microglia Phenotypes, Especially Protective Microglia, as a New Therapeutic Target

A detailed classification of microglia may help in detecting new therapeutic targets. Since the 2010s, microglia have been categorized as “M1” (pro-inflammatory) and “M2” (anti-inflammatory) based on the concept established in macrophages [126]. However, in recent years, researchers have recommended using more nuanced tools (for example, proteomic, metabolomic, transcriptomic, morphological, and epigenetic tools) to investigate microglial functions because microglia have more complex responses than simply polarized responses, such as M1 and M2 [127]. The technologies that have been the basis for the research field of microglia by revealing heterogeneous cell states are scRNA-seq and snRNA-seq [128,129]. In 2017, scRNA-seq identified a group of microglia, called DAM, found around Aβ plaques which is a key microglial phenotype in AD [130]. DAMs upregulate genes, including *ApoE*, *Trem2*, *Itgax* (encoding integrin alpha X), *Igf1* (encoding insulin-like growth factor-l), *Lpl* (encoding lipoprotein lipase), and *Cst7* (encoding cystatin 7) [130]. DAMs are activated sequentially in two steps as TREM2-independent and -dependent pathways (Figure 3) [131]. The first step involves TREM2-independent downregulation of homeostatic markers, such as *P2ry12*, *Cx3cr1*, and *Tmem119*. The second activation state is TREM2-dependent. It is associated with the overexpression of genes involved in lipid metabolism and phagocytosis. DAM-like cells have also been observed in other animal models of diseases, such as amyotrophic lateral sclerosis and multiple sclerosis [132]. In addition, a recent study characterized a human AD microglia (HAMs) profile from the postmortem brain [133]. HAMs had profiles similar to those of DAM genes related to lipid transport and lysosomal biology, such as *ApoE*. However, it has also been reported that genes that are unchanged or not expressed in DAMs are upregulated in HAMs [133]. Therefore, the differences between human and animal models should be examined in more detail.

One of the highly upregulated genes in DAMs is *Itgax*, encoding integrin αX (ITGAX; also known as CD11c) [130]. It has been suggested that the subpopulations of microglia in AD, such as DAMs, reported in many recent studies reflect features of CD11c^+^ microglia identified more than a decade ago [134]. Only a small number of CD11c^+^ cells are present in mouse brain under normal conditions. In young mice, CD11c^+^/CD45^high^ macrophages are present in the juxta-vascular parenchyma and choroid plexus, and CD11c^+^/CD45^dim^ cells are observed in the brain parenchyma [135,136,137]. CD11c^+^/CD45^dim^ cells are considered to be microglia as the population expresses the microglial markers CX3CR1, IBA1, and CD11b, in combination with low expression of MHC class II [136]. It has been reported that CD11c^+^ cells in the brain increase in models of other CNS pathologies (experimental autoimmune encephalomyelitis (EAE) and kainic acid administration) and accumulate around injured sites [135,138]. In addition, recent research has shown that CD11c^+^ microglia in the spinal cord are essential for remission of nerve injury-induced chronic pain [139]. Thus, it appears that CD11c^+^ microglia play a protective role in CNS disorders [134].

In AD models, including 3xTg, APP/PS1, and 5xFAD, several studies have reported that CD11c^+^ cells accumulate around Aβ plaques [140,141,142]. It was also reported that in APP/PS1 mice over a period of 9–18 months, 23% of all IBA1^+^ cells around plaques were CD11c^+^ [140]. The transcriptomic signature of CD11c^+^ microglia indicate a suppressive/tolerizing effect of immune signaling by CD11c^+^ cells and an enhanced capacity for uptake and lysosomal degradation of Aβ by CD11c^+^ cells [140]. Interestingly, CD11c transcripts increase in APP/PS1 mice along with an age-related increase in plaque load but decrease in the later stages of AD [140]. These data suggest that CD11c^+^ microglia may exert a protective action against AD and that these microglial subsets could open new prospects as therapeutic targets for AD. CSF1 receptor stimulation promotes neuroprotection in CD11c^+^ microglia [143]. IL-34, a CSF1 receptor ligand, enhances the neuroprotective effects of microglia to attenuate Aβ neurotoxicity [144]. The development and safety evaluation of drugs that increase CD11c^+^ microglia are important in the future. 

## 8. Pharmacological or Diet Modulation on Microglial Responses in AD

As many studies have shown the importance of microglia in AD pathology, a pharmacological approach to modulate microglial function is currently being developed [145]. Glycogen synthase kinase-3β (GSK-3β), a serine/threonine kinase, is a powerful inflammation regulator [146]. GSK-3 inhibition using the small molecule inhibitor SB216763 reduced the LPS-induced elevated proinflammatory cytokine levels in rat glia-enriched cortical cultures [147]. Recent studies have shown that treatment with L807mts, a highly selective GSK-3 peptide derivative inhibitor, enhances the clearance of Aβ loads, reduces inflammation, and improves cognition in 5xFAD mice [148]. However, clinical studies have failed to identify effective and safe inhibitors of this kinase [149]. Targeting GSK3-β has many issues because it is ubiquitously expressed and relates to multiple cellular regulatory functions.

Current evidence suggests that peroxisome proliferator-activated receptor γ (PPAR-γ) activation may be useful for AD therapy. PPAR-γ is a nuclear receptor and a vital regulator of lipid metabolism, adipogenesis, and insulin sensitivity [150]. Pioglitazone, a PPAR-γ agonist, inhibits the production of pro-inflammatory cytokines in an LPS-activated rat microglial cell line [151]. Unfortunately, pioglitazone did not delay the onset of mild cognitive impairment in a clinical study (TOMMORROW: a prognostic biomarker study and a phase three, randomized, double-blind, and placebo-controlled trial) [152]. A phase IIa study provided some limited indications of T3D-959, a PPAR-γ, and PPAR-δ improving cognition [153]. The complex role of PPAR, including the metabolism of Aβ, requires detailed study.

Furthermore, foods and nutrients consumed daily may also modulate microglia [145]. Genistein, an isoflavone found in soy, exerts anti-inflammatory effects on LPS-stimulated BV-2 microglia [154]. Genistein improves microglial morphology and neuron–microglia contact in a mouse model of social defeat stress [155]. Clinical research has also indicated that genistein may play a therapeutic role in delaying AD onset [156]. Lycopene, highly concentrated in tomatoes, reduces cyclooxygenase-2 expression in BV-2 microglial cell activation after LPS treatment in vitro [157]. Moreover, lycopene attenuates LPS-induced cognitive impairments and amyloidogenesis by mediating neuroinflammation and oxidative stress in mice [158]. However, of the four human studies investigating circulating lycopene and preexisting dementia, only one reported significant association between high AD mortality rate and low circulating lycopene level [159].

As indicated above, several drugs and foods can be used for modulating microglia; however, these drugs and foods have diverse effects on different cells. Therefore, selective therapeutic targets for the treatment of microglial diseases should be identified. Further research on these candidate drugs, including changes in intracellular signaling and microglial phenotypes, will lead to the development of new therapeutics for AD.

## 9. Future Perspectives

As stated in this review, accumulating research has provided evidence that microglia contribute significantly to the pathogenesis of AD. The recent identification of microglial phenotypes by single-cell analysis will accelerate the investigation of various microglial functions in AD. Although there have been many studies on the advanced stages of AD, there is still a lack of research on the pre-onset and early stages of AD. Early detection and treatment of AD are desirable as there is no curative therapy to halt or reverse the disease. Clarifying when and how the characteristic states of microglia (e.g., DAMs and CD11c^+^) appear in the early phase of AD is of particular importance. The differences in microglial reactivity between animal models and patients with AD also need to be clarified. Further studies are needed to elucidate the molecular mechanisms and signaling pathways. For example, how TREM2 interacts with other molecules to regulate Aβ phagocytosis and neuroinflammation should be clarified. Furthermore, some individuals retain cognitive function despite high levels of AD pathology, including Aβ deposition, called ‘AD resilience’ [160]. As many risk factors for AD are expressed in microglia, there is a high possibility that microglia are involved in AD resilience. Indeed, high levels of sTREM2 in the CSF of patients have been shown to be associated with a slower decline in memory and cognition [161]. It would be interesting to further investigate the relationship between microglia and AD resilience. 

Another important point for future research is to generate AD models that mimic the pathology of sporadic AD patients. One of the reasons why many potential treatments for AD have not been validated in clinical trials may be the limitations of animal models. Genetic AD mouse models are widely used as wild-type mice do not develop senile plaques or neurofibrillary tangles at any age [162]. Although less than 5% of AD cases are caused by a single genetic variant that is passed on to family members, most AD mouse models to date have been generated based on familial AD-related genetic mutations, such as the Swedish mutation (KM670/671NL) and the Indiana mutation (V717F). Thus, increased densities and morphological changes of microglia have been reported in the AD mouse models presented in this review (*App*^N-L-GF^, Tg2576, J20, APP/PS1, 5xFAD, and 3xTg mice) [163,164,165,166,167,168]; however, an AD mouse model that induces deposition of both Aβ and tau, as observed in sporadic AD patients, has not yet been established. While the Aβ protein sequence is highly conserved across mammals, the microtubule-associated protein tau (MAPT) sequence appears less conserved with substantial variations in the presence of tau isoforms [169]. To decipher data from in vivo experiments, it is necessary to understand the characteristics and shortcomings of each animal model. A greater understanding of the pathogenesis of Aβ and tau deposition will help to develop more accurate disease models.

Research on the interaction between AD and other diseases is also important. This review discusses the association between chronic pain and Alzheimer’s disease (AD). For example, recent clinical research has demonstrated that the risk of dementia is higher in participants with knee pain; however, the underlying mechanism is unknown. In patients with knee pain and in animal models, studying AD-related biomarkers or microglia-related factors in brain regions associated with AD pathogenesis will help further elucidate the pathological mechanisms. Elucidating these issues will provide great advances in the understanding of microglia and open new avenues for the development of therapeutics for AD.

## Figures and Tables

**Figure 1 cells-12-01207-f001:**
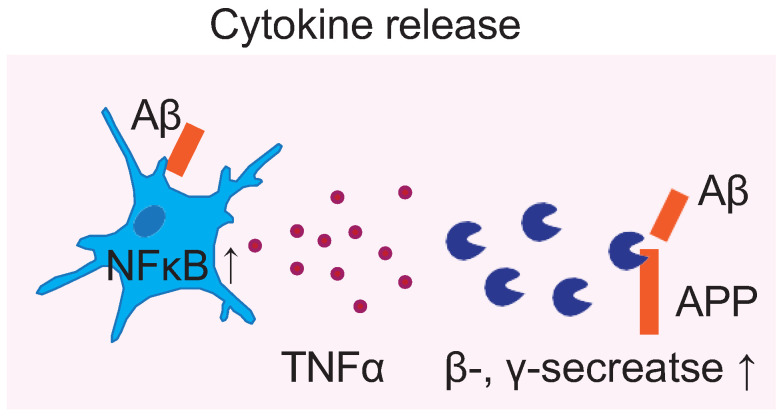
Various responses of microglia to amyloid β. Cytokine release: Aβ directly stimulates TNF-α production in microglia by activating the transcription factor NF-κB. TNF-α increases the expression of β- and γ-secretase which generate Aβ from APP.

**Figure 2 cells-12-01207-f002:**
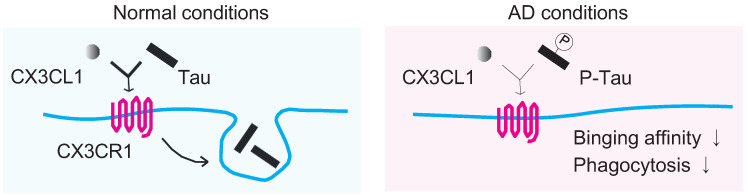
Phagocytosis: The CX3CL1/CX3CR1 axis plays a key role in the phagocytosis of tau by microglia and is affected as AD progresses. S396 Tau phosphorylation decreases the binding affinity of this protein to CX3CR1.

**Figure 3 cells-12-01207-f003:**
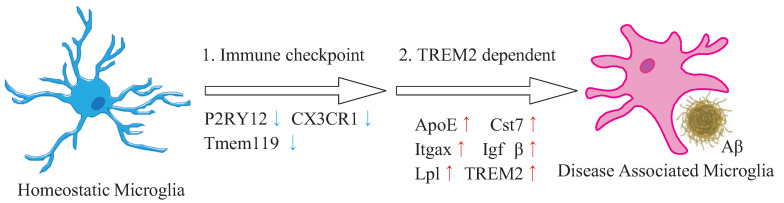
DAMs are activated sequentially by two steps. Schematic illustration showing the two steps by which microglia switch from homeostatic to DAMs. The first step: TREM2-independent downregulation of homeostatic markers such as P2RY12, CX3CR1, and Tmem119. The second step: TREM2-dependent overexpression of genes involved in phagocytosis and lipid metabolism. Arrows indicate up- or down-regulation of the gene.

## Data Availability

Not applicable.

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
