# Peer review of "The Functions and Phenotypes of Microglia in Alzheimer’s Disease"

_cells, 2023, doi:10.3390/cells12081207_

Round 1

Reviewer 1 Report

In the present review, the Authors discuss the role of microglia in AD. The manuscript is very interesting and informative. However, there are some points of concern, which should be properly addressed to further improve the quality of the manuscript.

Major points:
1. In subchapter 3, in lines 97-98 the Authors described the effect of Amyloid Beta on
microglial responses. However, in Figure 1 the Authors show also the effect of Tau
on the microglia responses. I suggest breaking figure 1 into two separate ones describing
the effect of AB (Figure 1) and Tau (Figure 2) on the microglial response.
2. Figure 2 I suggest put in section 4: Microglia responses to tau tangles.
In addition, I propose to present it in normal and pathological (AD) condition.
3. I suggest removing section 5: Mouse models of AD.
4. In line 257 explain the abbreviation BBB.
5. A section showing the effects from the pharmacological/ diet modulation
on microglial responses in AD should be included.  

Author Response

Comment 1: In subchapter 3, in lines 97-98 the Authors described the effect of Amyloid Beta on microglial responses. However, in Figure 1 the Authors show also the effect of Tau on the microglia responses. I suggest breaking figure 1 into two separate ones describing the effect of AB (Figure 1) and Tau (Figure 2) on the microglial response.

Reply: We thank this helpful comment. According to the comment, we have separated Figure 1 (Aβ) and Figure 2 (Tau).

Comment 2: Figure 2 I suggest put in section 4: Microglia responses to tau tangles. In addition, I propose to present it in normal and pathological (AD) condition.

Reply: According to this comment, we have put the Figure 2 in section 4. Now it shows AD conditions with decreased phagocytosis (page 6, lines 263).

Comment 3: I suggest removing section 5: Mouse models of AD.

Reply: We thank your comment. The description of mouse models has been shortened and moved to another chapter “Future perspectives” (page 10, line 443-460). As another reviewer also mentioned about the section5 “To focus only on animal models associated with microglial activation in AD”, we added the information about microglial activation (page 10, lines 450-453).

Comment 4: In line 257 explain the abbreviation BBB.

Reply: We appreciate your suggestion. According to the comment, we have explained the abbreviation (page 3, lines 112).

Comment 5: A section showing the effects from the pharmacological/ diet modulation on microglial responses in AD should be included.

Reply: We appreciate your helpful comment. According to the comment, we have provided a new section “The pharmacological or diet modulation on microglial responses in AD” (page 9, lines 385-422). We explained the effects of glycogen synthase kinase-3β, peroxisome proliferator-activated receptor γ, the soy isoflavone genistein, lycopene.

Reviewer 2 Report

The review elucidates the role of microglia in AD.

The authors introduced microglia in general, activation of microglia in response to amyloid beta dan Tau proteins, animal (mice) models of AD

My comments are as follows:

1. To convert the review into a systematic review

2. To focus only on animal models associated with microglial activation in AD

3. Section 8 can be further clarified with subheading titled: Expression of AD risk gene in microglia 

4. The role of microglial phenotypes in AD

5. Missing gap of studies / strong justification to conduct the "literature review". Some of the recent articles in which the contents are somehow similar to this review

https://www.frontiersin.org/articles/10.3389/fnagi.2019.00233/full

https://www.frontiersin.org/articles/10.3389/fncel.2021.749587/full

https://jneuroinflammation.biomedcentral.com/articles/10.1186/s12974-022-02580-1

https://www.sciencedirect.com/science/article/pii/S0022283619300646

6. To elaborate the gap of studies. Otherwise the current review may seem redundant

7. The review can be more focused to the following aspect:

i. The role of microglia in the AD pathogenesis - Neuroinflammation response

ii. Development of potential biomarkers for microglial activity in AD

iii. Targeting microglia for therapeutic development in AD

Author Response

Comment 1: To convert the review into a systematic review

Reply: For a systematic literature introduction we have replaced chapters and revised the proofreading of the review. As we mentioned in introduction (page 1, line41-46), in this review, we have summarized the risk genes expressed in microglia and the response of microglia to Aβ and tau, to argue the importance of microglia in AD. Moreover, we have summarized the relationship between chronic pain, microglia, and AD to facilitate a comprehensive understanding of AD, including its complications. Microglial phenotypes and candidate drugs have also been discussed to explore future therapeutic targets for AD.

Comment 2: To focus only on animal models associated with microglial activation in AD

Reply: We appreciate your helpful comment. We added the information about microglial activation (page 10, lines 450-453).

Comment 3: Section 8 can be further clarified with subheading titled: Expression of AD risk gene in microglia

Reply: According to this comment, we have changed the title (page 3, lines 99).

Comment 4: The role of microglial phenotypes in AD

Reply: We thank for your suggestion. We explained microglia phenotypes, especially protective microglia in chapter 7 (page 7, line 325). We have additionally described interleukins that may alter microglia to a protective phenotype; “CSF1 receptor stimulation promotes neuroprotection in CD11c+ microglia [143]. IL-34, a CSF1 receptor ligand, enhances the neuroprotective effects of microglia to attenuate Aβ neurotoxicity [144]. The development and safety evaluation of drugs that increase CD11c+ microglia are important in the future.” (page 9, lines 379-382)

Comment 5, 6: Missing gap of studies / strong justification to conduct the "literature review". Some of the recent articles in which the contents are somehow similar to this review. To elaborate the gap of studies. Otherwise the current review may seem redundant.

Reply: We appreciate your helpful comment. According to the comment, we have provided a new section (page 9, lines 384). We have explained previous studies of drugs and foods involved in microglial activity and described the gap of studies. We also discuss research gaps and issues in “Future perspectives” (page 9, line 424).

Compared to other reviews, we believe there is also novelty in the reference to the link between pain and AD involving microglia, described in section 6 “Pain is a risk factor for AD: possible involvement of microglial CNS inflammation” (page 6, line 269).

Comment 7: The review can be more focused to the following aspect:  i. The role of microglia in the AD pathogenesis Neuroinflammation response  ii. Development of potential biomarkers for microglial activity in AD  iii. Targeting microglia for therapeutic development in AD

Reply: We appreciate your helpful suggestion. We explained potential biomarkers for microglial activity (plasma-derived extracellular vesicles; page 2, lines 88-96). In addition, we added sTREM2 and CX3CL1 as potential biomarkers for AD (page 3, lines 146-150, page 6, line 254-261). To focus on inflammation and treatment, we have added a new chapter, describing the candidate drugs and their future perspectives (page 9, line 385-422).

Round 2

Reviewer 1 Report

Dear Authors, I have no more comments.

Reviewer 2 Report

-